# Failure of the Brain Glucagon-Like Peptide-1-Mediated Control of Intestinal Redox Homeostasis in a Rat Model of Sporadic Alzheimer’s Disease

**DOI:** 10.3390/antiox10071118

**Published:** 2021-07-13

**Authors:** Jan Homolak, Ana Babic Perhoc, Ana Knezovic, Jelena Osmanovic Barilar, Melita Salkovic-Petrisic

**Affiliations:** 1Department of Pharmacology, University of Zagreb School of Medicine, 10 000 Zagreb, Croatia; ana.babic@mef.hr (A.B.P.); ana.knezovic@mef.hr (A.K.); jelena.osmanovic@mef.hr (J.O.B.); melitas@mef.hr (M.S.-P.); 2Croatian Institute for Brain Research, University of Zagreb School of Medicine, 10 000 Zagreb, Croatia

**Keywords:** GLP-1, streptozotocin, Alzheimer’s disease, oxidative stress, brain-gut axis, redox homeostasis

## Abstract

The gastrointestinal system may be involved in the etiopathogenesis of the insulin-resistant brain state (IRBS) and Alzheimer’s disease (AD). Gastrointestinal hormone glucagon-like peptide-1 (GLP-1) is being explored as a potential therapy as activation of brain GLP-1 receptors (GLP-1R) exerts neuroprotection and controls peripheral metabolism. Intracerebroventricular administration of streptozotocin (STZ-icv) is used to model IRBS and GLP-1 dyshomeostasis seems to be involved in the development of neuropathological changes. The aim was to explore (i) gastrointestinal homeostasis in the STZ-icv model (ii) assess whether the brain GLP-1 is involved in the regulation of gastrointestinal redox homeostasis and (iii) analyze whether brain-gut GLP-1 axis is functional in the STZ-icv animals. Acute intracerebroventricular treatment with exendin-3(9-39)amide was used for pharmacological inhibition of brain GLP-1R in the control and STZ-icv rats, and oxidative stress was assessed in plasma, duodenum and ileum. Acute inhibition of brain GLP-1R increased plasma oxidative stress. TBARS were increased, and low molecular weight thiols (LMWT), protein sulfhydryls (SH), and superoxide dismutase (SOD) were decreased in the duodenum, but not in the ileum of the controls. In the STZ-icv, TBARS and CAT were increased, LMWT and SH were decreased at baseline, and no further increment of oxidative stress was observed upon central GLP-1R inhibition. The presented results indicate that (i) oxidative stress is increased in the duodenum of the STZ-icv rat model of AD, (ii) brain GLP-1R signaling is involved in systemic redox regulation, (iii) brain-gut GLP-1 axis regulates duodenal, but not ileal redox homeostasis, and iv) brain-gut GLP-1 axis is dysfunctional in the STZ-icv model.

## 1. Introduction

Alzheimer’s disease (AD) is the most common type of dementia characterized by progressive neurodegeneration and the development of cognitive deficits. Etiopathogenesis of the disease is yet to be elucidated, with the exception of a small fraction of cases in which Mendelian inheritance of num, but not in the ileum of the controls. In the STZ-icv, TBARS and CAT were increased, LMWT amyloid precursor protein (APP), presenilin-1 (PSEN1) and presenilin2 (PSEN2) is believed to be a causative factor [1]. Many hypotheses have been proposed over the years to explain early molecular mechanisms responsible for the development of the disease. Since its proposal in 1992 [2], the amyloid cascade hypothesis has dominated the field. Nevertheless, the hypothesis is being increasingly criticized as none of the amyloidocentric drugs tested so far reached the predetermined primary endpoints [3,4]. Consequently, other hypotheses are being explored to provide novel therapeutic and diagnostic solutions and address the unmet needs of the increasing burden of sporadic AD (sAD). The metabolic hypothesis of AD is gaining increasing attention as insulin-resistant brain state (IRBS) is recognized as an important etiopathogenetic factor [5], and insulin resistance provides a common link between other hypotheses of AD [6].

Following the discovery of insulin and insulin receptors (IR) in the brain [7,8], and their abundance in brain regions involved in the regulation of cognitive function and metabolism [9], Hoyer and colleagues proposed dysfunctional brain insulin signaling might be involved in the development of the metabolic dyshomeostasis recognized as an important early molecular event preceding neuropathological changes in AD [10,11]. Early clinical findings supported the hypothesis. In one of the first clinical studies on the topic, Bucht et al. reported increased insulin levels during the oral glucose tolerance test in patients diagnosed with AD in comparison with hospitalized control patients [12]. Many studies followed providing accumulating evidence regarding the association of both central and peripheral metabolic dysfunction with AD. Excess body weight, obesity and metabolic syndrome during middle-age have all been recognized as risk factors for the development of AD [13], and diagnosis of type 2 diabetes mellitus (T2DM) has been associated with two times greater risk for the development of AD in the prospective population-based Rotterdam cohort [14]. The observed risk was even more pronounced in a subpopulation with a more advanced stage of T2DM as patients using exogenous insulin were found to be at a four-fold greater risk of AD in comparison with the controls [14]. More than three decades after the first metabolic hypotheses [11], IRBS is now recognized as an important etiopathogenetic factor and pharmacological target for AD [5]. Consequently, animal models of IRBS became increasingly relevant in the context of preclinical AD research, and antidiabetic drugs are emerging as an attractive therapeutic option for targeting IRBS in neurodegeneration [15].

Hoyer and colleagues [16] and Lackovic and Salkovic [17] introduced intracerebroventricular treatment with low-dose streptozotocin (STZ-icv) for modeling IRBS and sporadic AD-related changes in rodents. Streptozotocin (STZ) is a nitrosourea compound used for modeling type 1 [18] and type 2 [19] diabetes mellitus in experimental animals when administered parenterally. When administered intracerebroventricularly in a low dose, streptozotocin causes brain oxidative stress [20], mitochondrial dysfunction [21], neuroinflammation [22,23], cholinergic deficits [24], metabolic dysregulation [25], insulin system dysfunction [26], glucose hypometabolism [27], pathological accumulation of amyloid β [28] and hyperphosphorylated tau protein [26,29]. Most importantly, neuropathological changes are accompanied by the progressive development of cognitive deficits following the administration of STZ [26,30]. Consequently, considering the model recapitulates many important pathophysiological phenomena consistently and relatively expeditiously (cognitive deficits are standardly present 1 month after model induction), the STZ-icv treatment has been widely used for modeling different pathophysiological aspects of AD.

Following the recognition of the role of both central and peripheral insulin resistance as risk factors for the development of sAD, repurposing antidiabetic drugs emerged as an attractive potential therapeutic strategy for targeting IRBS [31,32]. So far, encouraging preclinical [33,34] and clinical [35,36] studies reported protective effects of intranasal insulin, and there is evidence that other antidiabetic drugs such as agonists of peroxisome proliferator-activated receptors γ (PPARγ) [37,38,39], metformin [40] or inhibitors of dipeptidyl peptidase-4 (DPP-4) [41] might also be useful.

Agonists of the glucagon-like peptide-1 (GLP-1) receptors are another important class of antidiabetic drugs extensively studied in the context of neurodegeneration for their anti-inflammatory, neuroprotective [42] and insulin-sensitizing properties [43]. Even though the most well-known effect of GLP-1 is potentiation of pancreatic insulin secretion following meal ingestion, numerous extrapancreatic effects have been reported [44]. Both GLP-1 receptors (GLP-1R) and GLP-1 are expressed in the CNS [42,44], and most of the physiological actions of GLP-1 seem to, at least partially, rely on GLP-1 signaling in the brain [44]. In the brain, GLP-1R are primarily expressed in neurons, especially in the hippocampus, neocortex and in the Purkinje cells of the cerebellum, and glial cell expression of GLP-1R can be induced by neuroinflammation [42]. Even though the exact mechanisms responsible for the neuroprotective effects of GLP-1 are still being explored, it has been proposed that GLP-1 acts as a classic growth factor activating transcription of genes related to cell growth, enhanced metabolism, inhibition of apoptosis and reduction of inflammation [42]. Furthermore, the ability of GLP-1 to suppress oxidative stress has been proposed as a possible mediator of neuroprotection demonstrated in a wide variety of in vitro and in vivo models [45,46,47,48]. Bidirectional regulation of GLP-1 and other growth factors has also been reported. In this context, the association of GLP-1 and insulin-like growth factor (IGF) signaling [44] is especially interesting as it has been shown that GLP-1 increases the expression of IGF-1 receptor (IGF-1R) [49], and that knockdown of IGF-1R diminishes antiapoptotic effects of GLP-1 in the periphery [50]. Furthermore, RNA silencing or antisera-induced reduction of IGF-2 was able to alleviate the protective antiapoptotic effect of GLP-1 in MIN-6 cells, and cells from IGF-1 receptor knockout mice are insensitive to GLP-1-induced increase in β cell proliferation [44,49]. It is still unknown whether this close functional relationship of GLP-1 and IGF signaling is specific for pancreatic cells, and some questions have been brought up [51] regarding the methodology of the observed findings reported in [49]. Nevertheless, other findings suggest that this might be true as insulin and IGF-1 signaling pathways are closely related and GLP-1 mimetics have been reported to re-sensitize insulin signaling in the brain in different animal models of AD [52,53]. If confirmed, the existence of such a biological relationship would further reinforce the importance of GLP-1 and GLP-1 agonists in the context of IRBS and neurodegeneration as brain IR and IGF-1R use the same intracellular signaling cascade, and they are often present in a heterodimerized form [54]. Furthermore, the development of the resistance to both insulin and IGF-1 has been recognized as an overlapping phenomenon implicated in IRBS and AD [55].

The gastrointestinal (GI) tract is emerging as an overlooked player involved in the pathogenesis of AD. Accumulating evidence suggests gut microbiota might be involved both in the etiopathogenesis and the modulation of the course of the disease by increasing permeability of the gut and blood-brain barrier, secreting a large amount of amyloid and pro-inflammatory molecules [56]. Furthermore, recent mechanistic experiments demonstrated that intra-gastrointestinal administration of Aβ oligomers can perturb enteric function, induce cerebral amyloidosis following retrograde transport through the vagus, and promote the development of cognitive impairments in mice [57]. A similar pattern of pathophysiological events has been reported for other proteins implicated in neurodegeneration such as α-synuclein indicating a common biological mechanism [58]. The GI tract is also the main source of GLP-1. GLP-1 is produced in enteroendocrine L cells that are in direct contact with luminal nutrients [44,59]. Even though L cells are predominantly located in the ileum and colon, recent evidence indicates duodenal L cells might also play an important role in GLP-1 secretion [59]. Meal ingestion stimulates a biphasic secretion of GLP-1 that seems to depend on yet unresolved mechanisms involving glucose-dependent insulinotropic peptide action on the cholinergic fibers of the vagus [59]. Brain GLP-1 signaling depends heavily on the GI GLP-1R as dorsal vagal complex, the primary cerebral pre-proglucagon expressing site in the CNS, receives regulatory visceral sensory inputs from gastro-duodenal neurons [60]. Consequently, dysfunction of secretion of gut GLP-1, or failure to trigger its central secretion might induce pathophysiological milieu favoring neurodegenerative processes. Interestingly, we have found plasma concentration of the active fraction of GLP-1 to be reduced in a rat model of sporadic Alzheimer’s disease induced by intracerebroventricular streptozotocin (STZ-icv) three months following model induction—a period corresponding with the early stage of the disease [27]. Moreover, it has been reported that galactose stimulates the secretion of GLP-1 [61], and chronic oral galactose treatment can normalize plasma active GLP-1 in the STZ-icv rats [27]. In contrast to parenterally administered galactose often exploited for modelling age-related pathophysiology in rodents [62] chronic oral galactose treatment has been shown to both prevent and alleviate cognitive decline in the STZ-icv animals [27,63]. Even though possible mechanisms mediating both harmful and beneficial effects of galactose are still insufficiently explored [64], it has been proposed that restoration of GLP-1 secretion might be responsible for neuroprotective effects in the STZ-icv rats [27].

Based on the aforementioned, GLP-1 is emerging as an important player in neurodegeneration both in the context of its potential pathophysiological role and as an attractive pleiotropic therapeutic target [65] modulating IRBS, peripheral insulin resistance, accumulation of amyloid β, oxidative stress, neuronal cell proliferation and differentiation, apoptosis and synaptic plasticity [43,66,67,68,69,70]. Consequently, there is an increasing need to better understand the diverse physiological roles of GLP-1 that would then enable us to fully exploit its therapeutic potential. The present aim was to assess whether brain GLP-1 signaling is involved in the regulation of GI homeostasis utilizing acute pharmacological inhibition of brain GLP-1R. The existence of such physiological feedback loop might explain one potential pathway by which disruption of the GLP-1 system at the level of either brain or gut might generate a pathophysiological milieu favoring neurodegeneration. Furthermore, we were interested to see whether the brain-gut GLP-1 axis is preserved in the rat model of sAD given the previously reported perturbance of GLP-1 signaling [27].

## 2. Materials and Methods

### 2.1. Animals

Three-month-old male Wistar rats (*n* = 40) from the animal facility at the Department of Pharmacology (University of Zagreb School of Medicine) were included in the experiment. The animals were kept 2–3 per cage with a 7AM/7PM light-dark cycle, and standardized pellets and water available *ad libitum*. Humidity and temperature were in the range of 40–70% and 21–23 °C respectively. The bedding was changed twice per week.

### 2.2. Streptozotocin Treatment

The STZ-icv model was generated as described previously [23,27]. Briefly, rats were randomized to two groups and anesthetized with ketamine (70 mg/kg) and xylazine (7 mg/kg), the skin was surgically opened and the skull was trepanated bilaterally. Streptozotocin (1.5 mg/kg dissolved in 0.05 M citrate buffer, pH = 4.5) or vehicle was split into two equal doses and administered bilaterally (2 μL/ventricle) directly into the brain ventricles as first described by Noble et al. [71]. Freshly made STZ was used, and the treatment was delivered by a Hamilton microliter syringe with a custom-made stopper [72] at coordinates −1.5 mm posterior; ±1.5 mm lateral; +4 mm ventral from pia mater relative to bregma. The skin was sutured, and the same procedure was repeated after 48 h. Each animal in the STZ-icv group received a cumulative dose of 3 mg/kg streptozotocin.

### 2.3. Exendin-3(9-39) Amide Treatment and Tissue Collection

One month after the STZ-icv, animals from both the control (CTR; *n* = 20) and STZ-icv (STZ; *n* = 20) group were randomized to receive either saline or GLP-1R antagonist Exendin-3(9-39)amide (Ex-9) (Tocris Bioscience, Bristol, UK) (85 μg/kg dissolved in saline) by a single intracerebroventricular injection (CTR (*n* = 10); STZ (*n* = 10); CTR Ex-9(*n* = 10); STZ Ex-9 (*n* = 10)) The same procedure and coordinates were used as described for STZ-icv. 30 min after the treatment, 6 animals from each group were euthanized in general anesthesia and decapitated (the rest of the animals underwent the transcardial perfusion procedure). Proximal duodenum (post-gastric 2 cm) and distal ileum (pre-caecal 2 cm) were dissected and cleared from the surrounding tissue and luminal content in ice-cold phosphate-buffered saline (PBS). The tissue was snap-frozen in liquid nitrogen and stored at −80 °C. Afterward, the samples were homogenized on dry ice and subjected to three cycles of sonification (Microson Ultrasonic Cell 167 Disruptor XL, Misonix, SAD) in five volumes of lysis buffer containing 150 mM NaCl, 50 mM Tris-HCl pH 7.4, 1 mM EDTA, 1% Triton X-100, 1% sodium deoxycholate, 0.1% SDS, 1 mM PMSF, protease inhibitor cocktail (Merck, Burlington, USA) and phosphatase inhibitor (PhosSTOP, Roche, Switzerland) (pH 7.5) on ice. Homogenates were centrifuged for 10 min at RPM and 4 °C. The protein concentration of the supernatant for further analytical correction was measured utilizing the Lowry protein assay [73] and supernatants were stored at −80 °C until further analysis. Plasma was extracted from whole blood drawn from the retro-orbital sinus after centrifugation at 3600 RPM at 4 °C for 10 min in heparinized tubes (100 μL/sample).

### 2.4. Superoxide Dismutase Activity

Superoxide dismutase activity was measured by sample-mediated inhibition of 1,2,3-trihydroxybenzene (THB) autooxidation as described previously [64,74]. Briefly, 15 μL of 60 mM THB dissolved in 1 mM HCl was added to 1000 μL of 0.05 M Tris-HCl, and 1 mM Na_2_EDTA (pH 8.2), briefly vortexed and mixed with 10 μL of the sample. Absorbance increment was recorded at 325 nm for 300 s. Maximal THB autooxidation was measured with the same procedure omitting the sample. Autooxidation inhibition was the ratio of sample and reference sample difference at the endpoint and baseline absorbance values, ratiometrically corrected for tissue sample protein concentration. CamSpec M350 DoubleBeam UV-Visible Spectrophotometer (Cambridge, UK) was used.

### 2.5. Lipid Peroxidation

Lipid peroxidation was measured utilizing thiobarbituric acid reactive substances (TBARS) assay [64,75]. Briefly, 12 μL of tissue homogenate was mixed with 120 μL TBA-TCA reagent (0.375% thiobarbituric acid in 15% trichloroacetic acid) and 70 μL of ddH_2_O. Samples were incubated for 20 min in a heating block set at 95 °C in perforated microcentrifuge tubes. The complex of thiobarbituric acid and malondialdehyde was extracted in 220 μL n-butanol. The absorbance of the butanol fraction was analyzed at 540 nm in a 384-well plate using an Infinite F200 PRO multimodal microplate reader (Tecan, Switzerland). Predicted malondialdehyde (MDA) concentration was extracted from a linear model generated from a standard dilution curve prepared by dissolving MDA tetrabutylammonium stock in ddH_2_O.

### 2.6. Nitrocellulose Redox Permanganometry

Nitrocellulose redox permanganometry (NRP) was used for the determination of plasma and tissue reductive capacity as described in [76] and used in [64]. 1 μL of plasma was loaded onto the nitrocellulose membrane (Amersham Protran 0.45; GE Healthcare Life Sciences, Chicago, IL, USA) and left to dry out. Once dry the membrane was immersed in NRP reagent (0.2 g KMnO_4_ in 20 mL ddH_2_O) for 30 s. The reaction was terminated in dH_2_O, and trapped MnO_2_ precipitate was analyzed by densitometry of digitalized membranes in Fiji (NIH, USA). The same protocol was used for the assessment of tissue homogenates, however here the obtained values were ratiometrically corrected for respective sample protein concentration.

### 2.7. Low Molecular Weight Thiols and Protein Sulfhydryl Content

Low molecular weight thiols (LMWT) and protein sulfhydryl (SH) concentration were estimated by measuring the formation of 5-thio-2-nitrobenzoic acid (TNB) in a reaction between sulfhydryl groups and 5,5′-dithio-bis(2-nitrobenzoic acid) (DTNB) [64,75,77]. Briefly, 25 μL of tissue homogenate was incubated with 25 μL of 4% *w*/*v* sulfosalicylic acid for 1 h on ice and centrifuged for 10 min at RPM. 30 μL of the supernatant was transferred to separate wells for LMWT determination. The protein pellet was mixed with 35 μL of DTNB (4 mg/mL in 5% sodium citrate), left to react for 10 min and the supernatant absorbance was read at 405 nm using Infinite F200 PRO multimodal microplate reader (Tecan, Switzerland) to assess protein SH. The remaining supernatant was mixed with the same DTNB reagent, and its 405 nm absorbance was used for the assessment of LMWT. Both SH and LWMT concentration was calculated using a molar extinction coefficient of M^−1^cm^−1^.

### 2.8. Catalase Activity

Catalase activity in tissue homogenates was estimated indirectly from the H_2_O_2_ dissociation rate as proposed by Hadwan [78]. Briefly, 18 μL of tissue homogenate was placed in a 96-well plate. Sample background absorbance was checked at 450 nm. H_2_O_2_ concentration was determined by oxidation of cobalt (II) to cobalt (III) in the presence of bicarbonate ions by quantification of carbonato-cobaltate (III) complex ([Co(CO_3_)_3_]Co) at a 450 nm in t_0_ = 0 s (baseline) and t_1_ = 120 s (final). 10 mM H_2_O_2_ in 1xPBS (40 μL) was used as a substrate solution and Co (NO_3_)_2_ in hexametaphosphate and bicarbonate buffer as a reacting/stop solution [78]. Absorbance was measured with Infinite F200 PRO multimodal microplate reader (Tecan, Zürich, Switzerland). The concentration of H_2_O_2_ was determined from the model based on the standard curve obtained from a serial dilution of H_2_O_2_ in 1xPBS. Catalase activity was calculated from the difference in H_2_O_2_ concentration and was corrected for protein concentration and reaction time.

### 2.9. Data Analysis

Data were analyzed in R (4.0.2) in concordance with principles of reporting experimental data from animal studies [79] and communicating scientific evidence [80,81,82,83,84]. Sample size was chosen based on our previous experiments and no blinding was used throughout the experiment. Overall effects of treatment 1 (th1; intracerebroventricular citrate buffer vs. STZ) and treatment 2 (th2; intracerebroventricular saline vs. Ex-9) were first analyzed by linear regression using oxidative stress markers of interest as the dependent variables and th1 and th2 as independent variables. Subsequent analysis of STZ moderation of the Ex-9 effect was explored by including the th1:th2 interaction term. Model assumptions were checked using visual inspection of residual and fitted value plots. Differences of estimated marginal means or ratios for log-transformed dependent variables and respective 95% confidence intervals were reported for both models. The results were reported as raw data points (scatterplot) accompanied by boxplots graphically depicting data through their quartiles. Both the main effects models and models defined by the th1:th2 interaction term were visually reported as effects plots [79] with point estimates and confidence intervals of either differences or ratios of estimated marginal means plotted alongside respective contrasts tested. Interaction term *p*-values were reported alongside contrast estimates. Model outputs were reported in the text in the following format: contrast (e.g., STZ-CTR (indicating difference of estimated marginal means for the STZ-icv treated animals and the controls); overall/interaction (indicating whether the effect was estimated from the model without [overall], or with the interaction term [interaction]; main effects were described when appropriate (e.g., where the effect of central inhibition did not differ in the CTR and STZ animals)); CI: (e.g., 1.36–3.68 (indicating confidence interval for the effect to inform the reader about the strength of the effects and depict how precisely the effect has been estimated)[81,82]); t_df_ (indicating the t-statistic value with _df_ denoting degrees of freedom)[83]; p_raw_ = (the *p*-value obtained by the model).. Principal component analysis was used for dimensionality reduction. The same modeling approach for the individual oxidative stress parameters was used to assess the overall effect on the redox regulatory network by using the position of an individual animal in respect to the first principal component as the dependent variable, and treatments as the independent variables. Additional parameters related to statistical results are provided in Appendix A.

## 3. Results

### 3.1. Acute Pharmacological Inhibition of Endogenous GLP-1R Signaling in the Brain Induces Systemic Oxidative Stress

Acute inhibition of endogenous GLP-1R signaling in the brain with Ex-9 induced peripheral oxidative stress in both control animals and the rat model of sAD. Three markers of oxidative stress were examined—SOD, TBARS and NRP. Plasma SOD activity was reduced in the STZ-icv model in comparison with controls (STZ-CTR_overall_ CI: 1.36–3.68; t_36_: 4.41; p_raw_ = 0.0001 | STZ SAL-CTR SAL_interaction_ CI: 1.61–5.70; t_35_: 4.81; p_raw_ = 0.0002). Furthermore, a trend of reduction of the activity was observed upon inhibition of endogenous brain GLP-1, but only in the control animals (CTR Ex-9-CTR SAL_interaction_ CI: −3.83–0.26; t_35_: −2.35; p_raw_ = 0.11) (Figure 1A,D). Plasma lipid peroxidation end products inferred from TBARS concentration suggested acute inhibition of endogenous brain GLP-1 reduces lipid peroxidation in both control and STZ-icv animals (Ex-9-SAL_overall_ CI: −0.02–−0.003; t_36_: −2.72; p_raw_ = 0.01)(Figure 1B,E). The overall reductive capacity of plasma measured by NRP revealed weakened antioxidant capacity in STZ-icv animals (CTR-STZ_overall_ CI: 1.13×10^6^–4.69 × 10^6^; t_36_: 3.31; p_raw_ = 0.002 | CTR SAL-STZ SAL_interaction_ CI: 1.59 × 10^5^–6.81 × 10^6^; t_35_: 2.83; p_raw_ = 0.037) and suggested acute pharmacological inhibition of brain GLP-1 signaling to reduce plasma reductive capacity (Ex-9-SAL_overall_ CI: −5.07 × 10^6^–−1.52 × 10^6^; t_36_: −3.76; p_raw_ = 0.0006) with the effect clearly observed in control animals (CTR Ex-9-CTR SAL_interaction_ CI: −7.20 × 10^6^–−5.47 × 10^5^; t_35_: −3.14; p_raw_ = 0.017) and a trend observed in the rat model of sAD (STZ Ex-9-STZ SAL_interaction_ CI: −6.10 × 10^6^–7.33 × 10^5^; t_35_: −2.12; p_raw_ = 0.167) (Figure 1C,F). Principal component analysis suggests the effect of inhibition of the brain GLP-1R on plasma NRP and SOD might be mediated by a common biological mechanism as they cluster together in the biplot (Figure 1G,H).

### 3.2. STZ-icv Rats Are Resistant to Gastrointestinal Redox Dyshomeostasis Induced by the Inhibition of Endogenous GLP-1R in the Brain

The effect of acute pharmacological inhibition of brain GLP-1R on the gastrointestinal redox homeostasis was examined by measuring total tissue reductive capacity (NRP), lipid peroxidation (TBARS), low molecular cellular antioxidants (LMWT), protein sulfhydryl groups (SH), hydrogen peroxide dissociation capacity (CAT) and the activity of ROS scavenger enzyme superoxide dismutase (SOD). All markers were examined in both duodenum and ileum to assess whether the observed effect was region-dependent. Total tissue reductive capacity was largely unchanged in both tissues (Figure 2A,G). Inhibition of brain GLP-1R increased lipid peroxidation in the duodenum and ileum of the control animals (duodenum CTR Ex-9-CTR SAL_interaction_ CI: 11.87–86.50; t_15_: 2.81; p_raw_ = 0.01 | ileum CTR Ex-9-CTR SAL_interaction_ CI: 15.12–65.56; t_19_: 3.35; p_raw_ = 0.003) (Figure 2B,H,M,R). In the STZ-icv group, inhibition of brain GLP-1R decreased lipid peroxidation in the duodenum (CTR Ex-9-CTR SAL_interaction_ CI: −82.33–−3.18; t_15_: −2.30; p_raw_ = 0.036) (Figure 2B,M), but produced no effect in the ileum (Figure 2H,R). Both LMWT (CTR Ex-9-CTR SAL_interaction_ CI: −16.38–−1.13; t_15_: −2.45; p_raw_ = 0.027) and SH (CTR Ex-9-CTR SAL_interaction_ CI: −12.22–−0.16; t_15_: -2.19; p_raw_ = 0.045) were reduced in the controls by the Ex-9 treatment in the duodenum, but no change was observed in the ileum (Figure 2C,D,I,J,N,O)). Both LMWT (CTR SAL-STZ SAL_interaction_ CI: 0.62–15.87; t_15_: 2.30; p_raw_ = 0.035) and SH (CTR SAL-STZ SAL_interaction_ CI: −0.42–11.64; t_15_: 1.98; p_raw_ = 0.066) were reduced in the duodenal tissue of STZ-icv rats, and no additional decrement was observed in the STZ-icv rats that also received Ex-9-icv (Figure 2C,D,N,O). This pattern was not reflected in the ileal homogenates (Figure 2I,J). Hydrogen peroxide dissociation rate demonstrated a trend of increment upon inhibition of brain GLP-1R in both duodenum (CTR Ex-9/CTR SAL_interaction_ CI: 0.58–10.06; t_15_: 1.31; p_raw_ = 0.210) and ileum (CTR Ex-9/CTR SAL_interaction_ CI: 0.71–3.72; t_19_: 1.23; p_raw_ = 0.235) in the controls, but while a similar response of the STZ-icv rats was observed in the ileum (STZ Ex-9/STZ SAL_interaction_ CI: 0.92–4.50; t_19_: 1.89; p_raw_ = 0.074), an inverse effect was detected in STZ-icv duodenal tissue (STZ Ex-9/STZ SAL_interaction_ CI: 0.04–0.80; t_15_: −2.45; p_raw_ = 0.027) (Figure 2E,K,P,S). Finally, SOD activity was decreased by the Ex-9-icv in the duodenum of the control animals (CTR Ex-9-CTR SAL_interaction_ CI: −0.14—−0.004; t_15_: −2.25; p_raw_ = 0.040), while the same effect was absent in the STZ-icv (Figure 2F,Q). No change of ileal SOD activity was induced by either treatment (Figure 2L). Principal component analysis dimensionality reduction suggests the observed changes of SH, LMWT and SOD reflect closely related biological mechanisms in both tissues evident from clustering of variable vectors in the biplot (Figure 3). Most of the observed oxidative stress-related changes were more pronounced in the duodenum, and the analysis of th:th interaction revealed differential responsiveness of duodenum to inhibition of brain GLP-1R in STZ-icv rats (Figure 2 and Figure 3).

## 4. Discussion

The presented results (**i**) suggest the involvement of brain GLP-1R in the regulation of systemic oxidative stress; (**ii**) provide the first evidence of the pathophysiological changes in the GI tract of the STZ-icv rat model of sAD; (**iii**) offer preliminary evidence of the involvement of brain GLP-1R in the regulation of GI redox homeostasis (and anatomical differences along the GI tract) and (**iv**) indicate dysfunction of the brain-gut GLP-1 axis in the STZ-icv rat model of sAD.

### 4.1. GLP-1 and Systemic Oxidative Stress

The protective effects of GLP-1 and GLP-1 agonists have been widely discussed in the context of oxidative stress [85,86,87], however, the contribution of endogenous GLP-1 to the regulation of systemic redox homeostasis by activation of brain GLP-1R is still not fully understood. Our results suggest that central GLP-1 might play a role in the systemic redox regulation as acute pharmacological inhibition of brain GLP-1R reduces reductive capacity measured by NRP (Figure 1C,F). Interestingly, plasma TBARS were reduced by the treatment both in the CTR and STZ groups (Figure 1B,E), however, the true meaning of this remains to be explored. Considering central GLP-1 is involved in the regulation of peripheral metabolism and blood flow [44], further analysis of other redox-related markers and the exploration of the effects on the liver and other peripheral organ systems involved in the brain-periphery GLP-1 axis could elucidate the mechanisms responsible for the observed effects. Reduced levels of SOD activity and diminished plasma reductive capacity observed in STZ-icv rats (Figure 1A,C,D,F) are in concordance with previous observations suggesting oxidative stress might play an important role in the development of neurodegenerative changes in the STZ-icv model [20,76,88].

### 4.2. Pathophysiological Involvement of the GI System in Animal Models of AD

Pathophysiological changes of the gut have been reported in different animal models of AD. In the Tg2576 mouse model of familial AD, dysregulation of gut homeostasis has been observed before the accumulation of brain Aβ [89]. Honarpisheh et al. analyzed the intestinal epithelial barrier of pre-symptomatic Tg2576 and found lower levels of mucus fucosylation and reduced expression of an important apical tight junction protein E-cadherin accompanied by an increased breach of gut bacteria through the epithelial barrier [89]. Increased intestinal permeability has been recognized as an important pathophysiological event that might promote enteric neuroinflammatory events and trigger neuroinflammation and neurodegeneration in the CNS [90]. In this context, failure of the GI homeostasis that emerges before both neuropathological and behavioral dysfunction suggests GI-related changes might be involved in the development of the AD-like phenotype. Interestingly, Tg2576 also suffers from an impaired absorptive capacity of vitamin B12 in the pre-symptomatic phase [89] and maintained B12 homeostasis is critical for the maintenance of white matter homeostasis, indicating other important mechanisms could also be involved in the GI-promoted neurodegeneration. Pathophysiological changes of the GI system have also been described in the transgenic models of AD such as 5xFAD, mThy1-hAβPP751, AβPP23 and TgCRND8 [91,92], however, to the best of our knowledge, this is the first evidence of GI-related pathophysiological changes in a non-transgenic model of AD. Non-transgenic models have been recognized as a valuable research tool for deciphering early molecular events related to AD so evidence of the involvement of the GI tract in both transgenic and non-transgenic animals might indicate important shared pathomechanisms. Furthermore, considering that the changes occur in the early stage of the disease [89,91], elucidation of the GI pathophysiology could provide foundations for the development of exciting new diagnostic opportunities and preventive strategies. As we have recently confirmed the presence of redox dysbalance in an independent cohort of STZ-icv animals and observed that oxidative stress was associated with morphometric changes of the GI barrier [93], this remains an important area of our future research. Functional consequences of the GI involvement should also be considered. For example, decreased absorption of drugs has been reported in the mouse model of familial AD [94]. In this context, an answer to the question of whether such changes also occur in non-transgenic models (e.g., the STZ-icv) could further increase the reliability and robustness of the model and improve the chances of developing new meaningful treatment strategies in the preclinical setting.

### 4.3. Brain-Gut GLP-1 Axis

Incretin effects of GLP-1 have been discovered in the 1980s, and they have been successfully exploited for pharmacological glucoregulation by the 2000s [95]. Nevertheless, the physiology of GLP-1 is still being actively explored, and many of its exciting roles are yet to be fully understood [44,96]. It has been proposed that brain GLP-1Rs are involved in the regulation of peripheral glucose homeostasis, fuel partitioning and monitoring energy levels to prepare the organism for the fasting that comes after a meal [44,60]. Many peripheral systems are involved in this regulation. Stimulation of brain GLP-1 with exendin-4 decreased insulin-induced muscle glucose uptake, increasing glucose availability for the replenishment of liver glycogen [97]. Furthermore, stimulation of brain GLP-1 potentiates insulin secretion from the pancreas but counteracts its vasodilatory effects in the femoral artery [98] possibly to prevent muscle glucose utilization and increase liver glycogen synthesis [60]. Conversely, intracerebroventricular administration of Ex-9 has been shown to increase muscle glucose utilization in an insulin-independent manner that requires an intact vagus [97]. Interestingly, it has been proposed that brain GLP-1-mediated peripheral glucoregulation is rendered dysfunctional due to chronic overstimulation in a diabetic state, and that chronic intracerebroventricular administration of Ex-9 can block the development of hyperinsulinemia and insulin resistance in mice fed with a high-fat diet [44,60,99]. The gut is the main source of GLP-1, however, it is still not clear whether it is also under the influence of the brain GLP-1, possibly via the GLP-1 feedback loop. The effect of acute pharmacological inhibition of the brain GLP-1R on the redox homeostasis of the gut (Figure 2 and Figure 3) suggests this might be the case. The brain-gut GLP-1 axis has been described primarily in the context of brain regulation of lipid absorption. Farr et al. demonstrated that brain GLP-1 is involved in the regulation of postprandial chylomicron secretion [100]. Central stimulation of GLP-1 receptors either with exendin-4 or indirectly with endogenous GLP-1 upon inhibition of brain DPP-IV reduced postprandial secretion of chylomicrons, and pre-treatment with intracerebroventricular Ex-9 annihilates the effect [100]. The effect seems to be dependent on the sympathetic nervous system outflow as it was not observed in the presence of adrenergic receptor antagonists, and it seems to be mediated by regulation of jejunal triglyceride availability and the activity of microsomal triglyceride transfer proteins [100]. The current understanding of the brain-gut GLP-1 axis-mediated regulation of the GI system is relatively humble, and at this moment, it is impossible to propose whether the regulation of redox homeostasis and lipid absorption are in any way associated. Nevertheless, further exploration of the brain-gut GLP-1 axis might provide valuable information not only for understanding the physiology of GLP-1 but also for understanding its pathophysiology, especially in AD as described in the next paragraph.

### 4.4. The Role of Brain-Gut GLP-1 Axis in AD?

As mentioned previously, a dysfunctional GLP-1 system has been observed in the STZ-icv rat model of sAD. Three months after the model induction, active GLP-1 plasma concentration was reduced in the STZ-icv animals [27]. Interestingly, chronic treatment with oral galactose restores plasma GLP-1 in the STZ-icv rats and induces the expression of hypothalamic GLP-1R [27]. The mechanism and exact temporal pattern of GLP-1 changes in the STZ-icv model are still unknown, but GLP-1 dysfunction might be involved both in the development and progression of AD-like neuropathology and behavioral dysfunction. It is currently believed that IRBS is the main mechanism by which STZ-icv generates neuropathological changes and GLP-1 could re-sensitize brain insulin signaling [52] possibly by acting on mechanisms underlying bidirectional regulation of GLP-1 and other growth factors in the brain (e.g., IGF-1[44]). The involvement of the brain-periphery GLP-1 axis cannot be excluded in this context either, as peripheral metabolic dysfunction has been recognized as an important risk factor for the development of AD [13]. This is probably best evident from animal models exploiting peripheral metabolic dysfunction for modeling neurodegeneration (e.g., high-fat diet-induced IRBS, peripheral streptozotocin-induced IRBS [101]). Finally, an important mechanism by which the brain-gut GLP-1 axis might be involved in the development and/or progression of neurodegeneration is related to its regulation of intestinal lipid absorption. Dysfunctional lipid absorption could not only exacerbate neurodegenerative processes by stimulating peripheral metabolic dyshomeostasis and inflammation, but also by regulating Aβ homeostasis in the intestine. It has been shown that dietary cholesterol affects the risk of AD, and that excess brain cholesterol increases the generation of amyloid peptides and the accumulation of amyloid plaques [102]. Furthermore, it has been reported that dietary cholesterol and saturated fats increase enterocyte amyloid synthesis [103] and it has been proposed that intestinally derived Aβ is a key regulator of chylomicron metabolism involved in the control of postprandial lipoproteins generated in a response to dietary fats [104]. Interestingly, subsequent studies by the same group suggested that saturated fats, and not dietary cholesterol stimulate intestinal Aβ and that dietary cholesterol might even exert a protective effect by reducing the amyloid burden [104] similar to what has been previously reported [105]. Even though intestinal Aβ is still not sufficiently explored, and the extent of its contribution to the systemic amyloid burden and transport is unknown, demonstration of retrograde transport of intestinal amyloid into the brain and subsequent seeding [57] indicate it might play an important role in AD. It is also possible that intestinal Aβ regulation and lipid absorption in the intestine are involved indirectly by affecting lipid transport mechanisms recognized as important risk factors for AD. For example, apolipoprotein E4 has been recognized as one of the most important risk factors for AD and an attractive therapeutic target as it is the most prevalent genetic risk factor affecting approximately 50% of patients [106]. Considering the close association of intestinal lipoprotein production and Aβ, the brain-gut GLP-1 axis could emerge as an important player in AD due to its involvement in the regulation of intestinal lipoprotein production.

The brain-gut GLP-1 axis also seems to be involved in the regulation of GI redox homeostasis, although the mechanistic explanation of the observed effect remains to be proposed. Intestinal redox homeostasis is involved in stem cell proliferation, enterocyte apoptosis, intestinal immune responses, nutrient digestion and absorption [107]. Furthermore, intestinal oxidative stress seems to play an important role in the onset and development of chronic gut inflammation. Depletion of the most abundant intracellular low molecular weight thiol glutathione (GSH) has been reported in Crohn’s disease and ulcerative colitis [108,109]. Furthermore, it has been shown that depletion of mucosal T cell intracellular GSH results in redox disequilibrium that favors the switch from tolerant to reactive state associated with intestinal inflammation [107,110]. Inflammation and disrupted homeostasis of the gut have serious implications for AD as increased permeability of the intestinal barrier can initiate and perpetuate both systemic and central inflammation [56,111] and this mechanism has been recognized in the context of dysbiosis-induced neuroinflammation [111,112]. Intestinal dyshomeostasis accompanied by barrier dysfunction leads to induction of pro-inflammatory cytokines that stimulate the expression of amyloid precursor protein and β-secretase resulting in accumulation of amyloid plaques [113]. Furthermore, microbial products (e.g., bacterial amyloids, lipopolysaccharide) activate pathogen-associated molecular pattern-sensitive signaling (e.g., via Toll-like receptor-2 or -4) that leads to activation of microglia and generation of proinflammatory cytokines such as tumor necrosis factor α and interleukin 1 β in the central nervous system [113,114].An association of inflammatory bowel disease and dementia has also been reported [115] providing further evidence of the possible involvement of gut homeostasis in the disorders of the CNS. Here, we demonstrated that single acute administration of Ex-9 in the brain can induce intestinal oxidative stress reflected by increased lipid peroxidation and decreased LMWT, and protein SH and SOD activity (Figure 2 and Figure 3) indicating a possible role of the brain GLP-1R in the regulation of gut redox homeostasis. Interestingly, although pronounced in the duodenum, the effect was largely absent in the ileum (Figure 2 and Figure 3) suggesting a physiological regulatory mechanism with a clear anatomical distinction in the GI tract. Implications of these findings are to be further explored, but the existence of brain GLP-1R-dependent regulation of homeostasis in the upper small intestine could be important in the context of the previously acknowledged role of brain GLP-1R in peripheral energy monitoring and fuel partitioning [44,60]. In the context of neurodegeneration, the existence of such a mechanism would further corroborate the importance of brain GLP-1 as a therapeutic target in neurodegeneration providing an additional mechanism by which normalization of central GLP-1 signaling might stop the vicious cycle of neuroinflammation and systemic inflammation (Figure 4). Furthermore, our recent observations suggest that glucose-dependent insulinotropic polypeptide (GIP) might act by a similar mechanism as acute inhibition of brain GIP receptors exerts similar effects on the duodenal redox homeostasis as has been shown here for GLP-1 [93]. These findings add another important perspective to the concept of utilizing dual incretin receptor agonists [53,116].

## 5. Conclusions

Our results provide the first evidence of pathophysiological changes in the GI system of the STZ-icv rat model of sAD indicating involvement of the GI tract might be a shared feature of different animal models of AD. Given recent evidence suggesting the involvement of the gut in the etiopathogenesis and progression of the disease in humans, further understanding of intestinal pathophysiology might provide critical information for understanding the biology of neurodegeneration and the development of new diagnostic and treatment strategies. Furthermore, perturbations of the GI homeostasis upon inhibition of brain GLP-1R indicate the existence of the brain-gut GLP-1 axis involved in the maintenance of redox balance in the upper small intestine. Implications of these findings remain to be fully explored, but brain-gut GLP-1 redox regulation might be involved in the peripheral fuel partitioning and could be important in the context of the development of chronic gut inflammation. In concordance with previous findings related to the dysfunctional GLP-1 system in the STZ-icv model, the reported results provide additional mechanistic insight into how the failure of the brain-gut GLP-1 axis might support the development of systemic and central inflammation in the rat model of sAD.

## 6. Limitations

Intracerebroventricular administration can perturb oxidative stress and glucose homeostasis and it has been shown that GLP-1 inhibits hyperglycemia-induced oxidative injury [85]. Even though we included appropriate controls to account for this, the obtained results may reflect mechanisms that are more relevant in the pathophysiological than in the physiological milieu (Appendix A).

Furthermore, in this manuscript we propose the existence of the brain-gut GLP-1 axis based on the results from acute experiments. Further research focused on chronic effects of brain GLP-1R inhibition on the homeostasis of the gut, as well as exploration of potential mechanisms responsible for mediation of the effects are needed to elucidate whether the observed is just a short-term phenomenon or represents an important pathophysiological process that could theoretically drive dyshomeostasis of the gut.

Another important limitation is that it was not possible to assess potential efflux of Ex-9 into the periphery upon intracerebroventricular administration. Consequently, the observed effects could theoretically reflect small concentrations of inhibitor reaching the intestines. Previous experiments utilizing intracerebroventricular administration of Ex-9 acknowledged this problem. For example, Kanoski et al. introduced additional controls coadministered with intraperitoneal Ex-9 in doses likely exceeding those that could have effluxed after intracerebroventricular administration [117]. Kanoski et al. used 25% and 75% of centrally administered doses [117] introducing a substantial safety margin considering previous work estimated approximately 17% of centrally administered Ex-9 reaches peripheral circulation using intranasal administration [118]. Similar design utilizing additional control animals co-administered with intraperitoneal Ex-9 may provide an insight into whether the effects observed here could have been caused by the effluxed Ex-9 that reached the intestines. Until such experiments are conducted, we only have indirect proof suggesting direct effect of effluxed Ex-9 is not likely the cause as (i) there is no difference in cyclic adenosine monophosphate in either duodenom or ileum between groups (unpublished data); (ii) there is no difference in the estimated expression of GLP-1R in either duodenom or periduodenal adipose tissue (unpublished data)—although this information has to be interpreted with caution as commercial GLP-1R antisera often provide unreliable results [119,120]; (iii) there is no difference in plasma or cerebrospinal fluid glucose concentration (Appendix A). Taken all together, we believe the abovementioned provides evidence that Ex-9 efflux is not the likely cause of the observed effects, however future experiments using a similar design as has been used by Kanoski et al. [117] will provide a final answer.

## Figures and Tables

**Figure 1 antioxidants-10-01118-f001:**
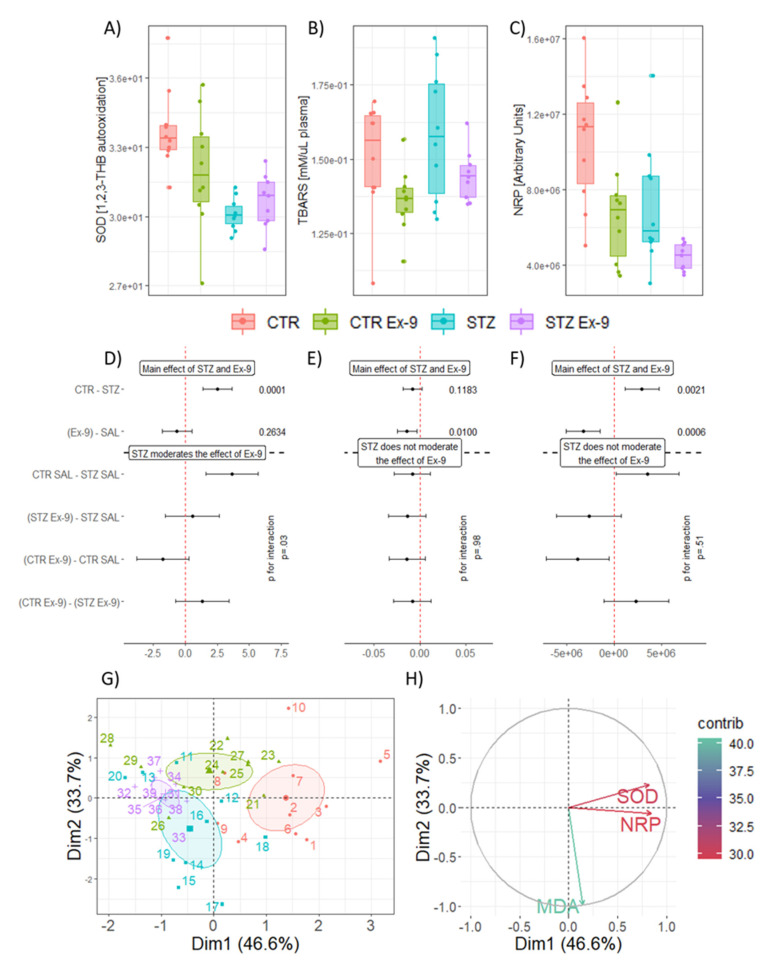
The effect of intracerebroventricular exendin-3(9–39)amide (icv-Ex-9) on plasma markers of oxidative stress in control rats and a rat model of sporadic Alzheimer’s disease induced by intracerebroventricular streptozotocin (STZ-icv). (**A**) Raw data of plasma SOD activity. (**B**) Raw data of plasma TBARS concentration indicating lipid peroxidation. (**C**) Raw data of plasma NRP indicating reductive capacity. (**D**) The effects of STZ and Ex-9 on plasma SOD activity reported as differences of estimated marginal means estimated from the main effects model (upper) or taking into account treatment interaction (lower). (**E**) The effects of STZ and Ex-9 on lipid peroxidation reported as differences of estimated marginal means estimated from the main effects model (upper) or taking into account treatment interaction (lower). (**F**) The effects of STZ and Ex-9 on total plasma reductive capacity reported as differences of estimated marginal means estimated from the main effects model (upper) or taking into account treatment interaction (lower). (**G**) Principal component analysis of oxidative stress-related variables in plasma. Individual animals are shown with respect to the biplot and shaded areas represent 95% confidence ellipses around group baricenters. (**H**) Contribution of variables in respect to the biplot. SOD—superoxide dismutase; TBARS—thiobarbituric acid reactive substances; NRP—nitrocellulose redox permanganometry. Dim1—1st principal component; Dim2—2nd principal component; e (×10^x^; scientific notation). CTR (*n* = 10); STZ (*n* = 10); CTR Ex-9 (*n* = 10); STZ Ex-9 (*n* = 8).

**Figure 2 antioxidants-10-01118-f002:**
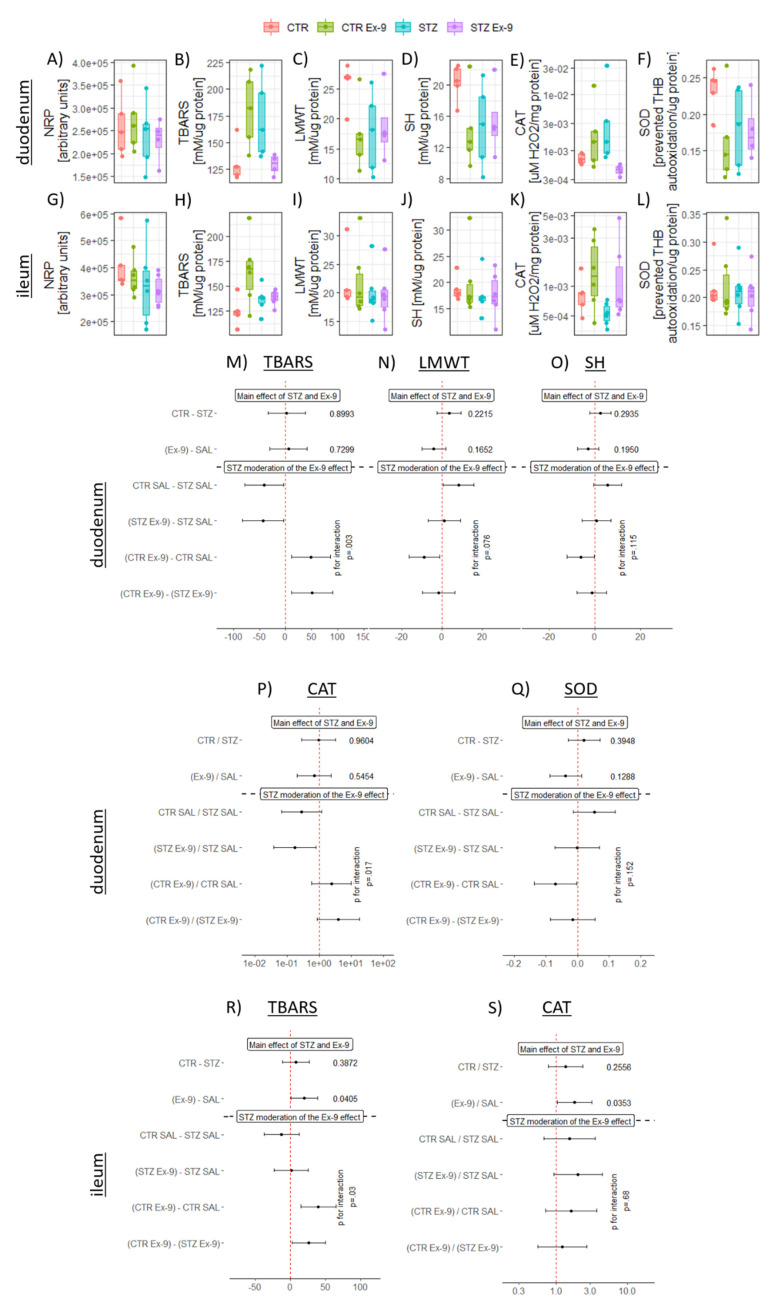
Oxidative stress markers in duodenal (**A**–**F**) and ileal (**G**–**L**) homogenates. (**A**) Overall reductive capacity in the duodenum. (**B**) Duodenal lipid peroxidation suggestive of qualitative th:th interaction. (**C**) Duodenal low molecular cellular antioxidants. (**D**) Duodenal total protein sulfhydryls. (**E**) Duodenal catalase activity suggestive of qualitative th:th interaction. (**F**) Duodenal superoxide dismutase activity. (**G**) Overall reductive capacity in the ileum. (**H**) Ileal lipid peroxidation. (**I**) Ileal low molecular cellular antioxidants. (**J)** Ileal total protein sulfhydryls. (**K**) Ileal catalase activity. (**L**) Ileal superoxide dismutase activity. Main effects and treatment interaction models for oxidative stress markers of interest in duodenal (M-Q) and ileal (R, S) homogenates. Differences of estimated marginal means and ratios of log-transformed variables marginal means are reported. *p*-values are reported for the main effects models and the interaction terms in the interaction models. Main effects and interaction models with duodenal (M) TBARS, (N) LMWT, (O) SH, (P) CAT and (Q) SOD used as dependent variables, and treatments used as independent variables. Main effects and interaction models with ileal (R) TBARS, and (S) CAT used as dependent variables and treatments as independent variables. CTR—control animals; CTR Ex-9—control animals treated intracerebroventricularly with Exendin-3(9–39)amide; STZ—rat model of sporadic Alzheimer’s disease induced by intracerebroventricular administration of streptozotocin (STZ-icv); STZ Ex-9—STZ-icv rats treated intracerebroventricularly with Exendin-3(9–39)amide; NRP—nitrocellulose redox permanganometry; TBARS—thiobarbituric acid reactive substances; LMWT—low molecular weight thiols; SH—protein sulfhydryls; CAT—catalase; SOD—superoxide dismutase; e (×10^x^; scientific notation).

**Figure 3 antioxidants-10-01118-f003:**
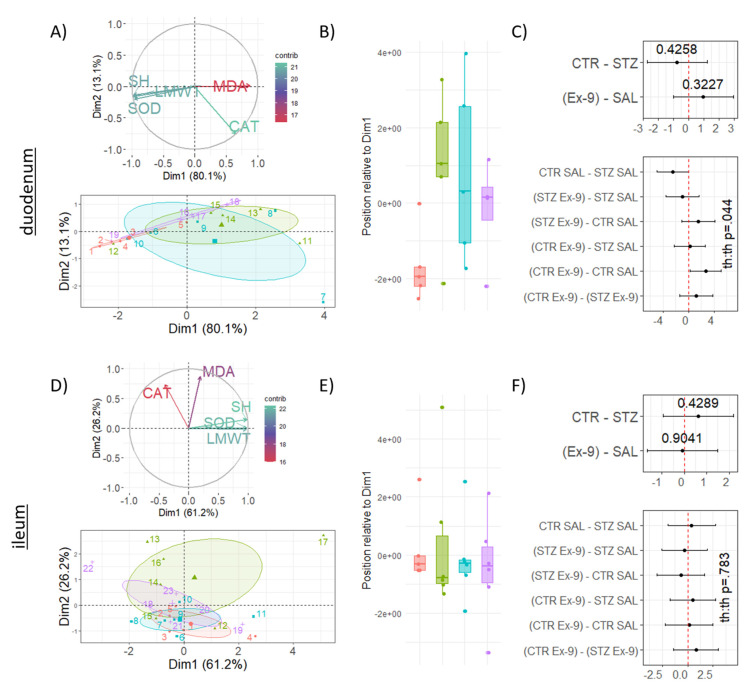
Dimensionality reduction of oxidative stress-related parameters in duodenal and ileal homogenates. (**A**) Principal component analysis of oxidative stress-related variables in the duodenum. The contribution of variables (upper) and individual animals (lower) are presented in respect to the biplot. Shaded areas represent 95% confidence ellipses around group baricenters. (**B**) Position of animals in respect to the 1st principal component. (**C**) Point estimates with corresponding confidence intervals from the main effects model (upper) or the treatment interaction model (lower). (**D**) Principal component analysis of oxidative stress-related variables in the ileum. The contribution of variables (upper) and individual animals (lower) are presented in respect to the biplot. Shaded areas represent 95% confidence ellipses around group baricenters. (**E**) Position of animals in respect to the 1st principal component. (**F**) Point estimates with corresponding confidence intervals from the main effects model (upper) or the treatment interaction model (lower). CTR—control animals; STZ—animals treated intracerebroventricularly with streptozotocin; Ex-9—animals treated intracerebroventricularly with Exendin 9–39; SAL—animals treated intracerebroventricularly with saline; TBARS—thiobarbituric acid reactive substances; LMWT—low molecular weight thiols; SH—protein sulfhydryls; CAT—catalase; SOD—superoxide dismutase. Dim.1—1st principal component; contrib—contribution; e (×10^x^; scientific notation).

**Figure 4 antioxidants-10-01118-f004:**
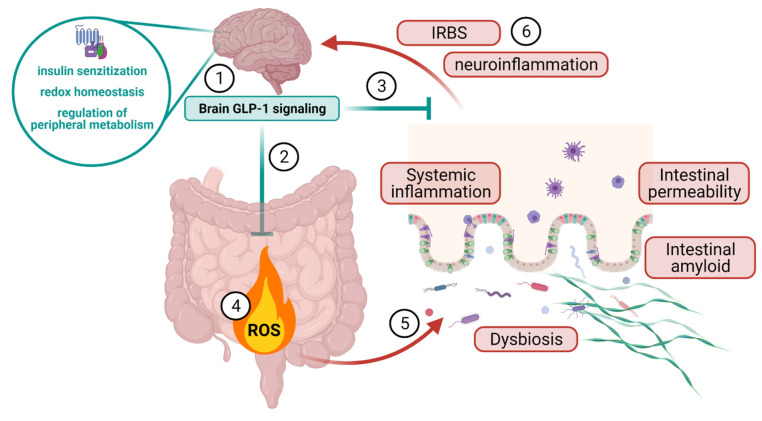
A schematic representation of the potential role of brain GLP-1 signaling in the control of intestinal, systemic and brain inflammation. Brain GLP-1 might control growth factor signaling in the brain (1), maintain the redox homeostasis of the gastrointestinal tract via the brain-gut axis (2) and control peripheral metabolism to regulate systemic inflammation (3). Failure of the brain GLP-1 signaling leads to dysfunctional redox homeostasis and generation of oxidative stress in the gut (4). Oxidative stress and loss of gastrointestinal homeostasis led to dysbiosis, accumulation of intestinal amyloid and increased epithelial permeability (5). Breach of microbiota, proinflammatory molecules and amyloid through the intestinal barrier induces systemic inflammation that in turn leads to the development of neuroinflammation and insulin-resistant brain state (6).

## Data Availability

A complete dataset and code used for the analysis is available upon request to the corresponding author.

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
