# Peer review of "Failure of the Brain Glucagon-Like Peptide-1-Mediated Control of Intestinal Redox Homeostasis in a Rat Model of Sporadic Alzheimer’s Disease"

_antioxidants, 2021, doi:10.3390/antiox10071118_

Round 1

Reviewer 1 Report

The paper by Homolak et al. provided novel evidence to support the existence of a brain-gut GLP1 axis mediating redox homeostasis in the upper small intestine. The manuscript is overall well-written. I am reviewing this manuscript from my expertise on nutritional metabolism and oxidative stress. The design of this study is one of my concerns since the authors are presenting data derived from a single administration of a brain GLP-1R inhibitor (to my understanding). The data presented here represent the acute effects of such administration but failed to establish or support the existence of bidirectional communication between the brain and the gut. Under “Limitations,” the authors acknowledge that ICV administration can affect the redox homeostasis, but could chronic administration of the drug produce the same effects, or are these effect effects just derive from the single administration? 

The effects they reported in the gut, are they exclusive of the gut? Furthermore, are there neurological/behavioral outputs or even brain tissue features that could have been assessed in the AD model in parallel?

Introduction: well written. Please revise: “mendelian” should be “Mendelian.”

Methods: page 4, line 162 statement regarding sample size should be mentioned in statistical section. It does not belong here.

Data analysis: statement regarding the excluded samples (page 6, line 260) is not relevant. There is no need for these details as long as the n is described in figure legends or in some way or form.

Overall, the results section must be improved. Authors use the following terms to describe their findings “slight reduction,” “can reduce,” but I don’t see an objective description of the data from a statistical perspective. Are there true differences? There is no mention of p-values, statistical differences, etc. When you mention “slight reduction,” is this a trend? Values look the same to me due to the variability. Please revise.

My comments above pertain to the figures as well. My statistical knowledge and expertise are limited, but the way data are presented is hard to follow; why not just present the results from the whole model, p-values for the main effects and interaction can be presented separately in a table. None of those p-values highlighted here (figure 1 D-F) are presented in results in such terms.

Page 8, line 321: “in contrast both LMWT and SH were reduced in the duodenal tissue of STZ-icv rats,”  compared to what?

Page 8, line 326 are there only trends? Please revise.

The authors failed to include the following in the discussion: how about inflammatory signaling? It is an important component in the pathophysiology of AD. How would you connect GI pathobiology to this aspect of  AD?

Page 14, line 463: authors talked about the role of oral galactose to restore plasma GLP-1 in the STZ-icv rats. If this is the case, how can you explain the galactose-induced models of neurodegeneration previously reported?

https://pubmed.ncbi.nlm.nih.gov/16555301/

What is the physiological or pathophysiological relevance of this research design considering the focus is a chronic neurodegenerative disease like Alzheimer’s? This is not addressed in the manuscript.

The effects they reported in the gut, are they exclusive of the gut? Furthermore, are there neurological/behavioral outputs or even brain tissue features that could have been assessed in the AD model in parallel?

Introduction: well written. Please revise: “mendelian” should be “Mendelian”.

Methods: page 4 line 162 statement regarding sample size should be mentioned in statistical section. It does not belong here.

Data analysis: statement regarding the excluded samples (page 6 line 260) is not relevant. There is no need for these details as long as the n is described in figure legends or in some way or form.

Overall, the results section must be improved. Authors use the following terms to describe their findings “slight reduction”, “can reduce” but I don’t see an objective description of the data from a statistical perspective. Are there true differences? There is no mention of p-values, statistical differences, etc. When you mention “slight reduction”, is this a trend? Values look the same to me due to the variability. Please revise.

My comments above pertain to the figures as well. My statistical knowledge and expertise are limited but the way data are presented is hard to follow, why not just present the results from the whole model, p-values for the main effects and interaction can be presented separately in a table. None of those p-values highlighted here (figure 1 D-F) are presented in results in such terms.

Page 8 line 321: “in contrast both LMWT and SH were reduced in the duodenal tissue of STZ-icv rats,”  compared to what?

Page 8 line 326 are there only trends? Please revise.

The authors failed to include the following in the discussion: how about inflammatory signaling? It is an important component in the pathophysiology of AD. How would you connect GI pathobiology to this aspect of  AD?

Page 14 line 463: authors talked about the role of oral galactose to restore plasma GLP-1 in the STZ-icv rats. If this is the case how can you explain the galactose-induced models of neurodegeneration previously reported?

https://pubmed.ncbi.nlm.nih.gov/16555301/

What is the physiological or pathophysiological relevance of this research design considering the focus is a chronic neurodegenerative disease like Alzheimer’s? This is not addressed in the manuscript.

Author Response

A point by point response to reviewers regarding „Failure of the brain glucagon-like peptide-1-mediated control of intestinal redox homeostasis in a rat model of sporadic Alzheimer’s disease“ by Homolak et al.

We wish to thank both the reviewers and the editors for their valuable time. We strongly the constructive comments will help us improve the manuscript.

Reviewer 1:

The paper by Homolak et al. provided novel evidence to support the existence of a brain-gut GLP1 axis mediating redox homeostasis in the upper small intestine. The manuscript is overall well-written. I am reviewing this manuscript from my expertise on nutritional metabolism and oxidative stress. The design of this study is one of my concerns since the authors are presenting data derived from a single administration of a brain GLP-1R inhibitor (to my understanding). The data presented here represent the acute effects of such administration but failed to establish or support the existence of bidirectional communication between the brain and the gut. Under “Limitations,” the authors acknowledge that ICV administration can affect the redox homeostasis, but could chronic administration of the drug produce the same effects, or are these effect effects just derive from the single administration?

We thank the reviewer for this comment. Based on current design we definietly cannot predict how long the observed effect would last, or whether it would be maintained in the scenario of prolonged inhibition of GLP-1 receptors in the brain. We strongly believe the effect has true biological meaning and that it is not an artifact as it was observed in an independent cohort of STZ-icv rats [1]. Nevertheless, new experiments are required to assess whether long-term inhibition would cause prolonged redox bisbalance in the gut. Testing this would also most likely require a different approach, as prolonged adinistration of the inhibitor would require a long-term placement of the microdialysis probe (inducing bias due to cerebral trauma), requiring understanding pharmacokinetics of the antagonist administared this way etc. Another approach would be to silent the receptors using a different strategy (e.g. siRNA, viral vectors, …). Several problems arise in this context as well because the effect would likely be compensated in the long run by other mechanisms working to perserve homeostasis. We understand that existence of the chronic effects of brain GLP-1R antagonism on the gut redox homeostasis would further strengthen the evidence for the existence of brain GLP-1R-dependent brain-to-gut communication, however the experiments that would provide reliable evidence in the context of chronic effects cannot currently be conducted by our team due to methodological reasons. We hope publication of the results from this manuscript will motivate other groups working in the field to examine the effects of chronic CNS GLP-1R inhibition on redox homeostasis of the gut.
We included a brief discussion of this important point in the Limitations section.

The effects they reported in the gut, are they exclusive of the gut? Furthermore, are there neurological/behavioral outputs or even brain tissue features that could have been assessed in the AD model in parallel?

The effects in the gut were our primary focus. Our previous experiments have shown that oral galactose exerts neuroprotective effects – preventing STZ-icv-induced cognitive deterioration if the therapy is started in parallel with intracerebroventricular administration of STZ [1], and alleviating STZ-icv-induced cognitive deficits in the design where it is started 1 month after model induction[2,3]. These findings were particularly interesting due to the fact that parenteral galactose is used for modeling aging in rodents indicating that gut, bypassed when galactose is used parenterally, might mediate at least some of the protective effects. Later, we have observed that plasma concentration of incretin hormones (particularly GLP-1) are increased in animals treated with galactose both acutely and chronically suggesting again that gut might be worth exploring in this context [3]. For this reason, we were particularly interested in the intestine (where the effects also seem to be region dependent as evident from the absence of the effect in the iluem).

In a different experiment utilizing a similar design, but focused on the inhibition of cerebral GIP receptor (central GIP inhibition seems to exert similar effects to what has been reported here for GLP-1) we also analyzed liver tissue redox homeostasis, however we observed no significant differences between the groups [manuscript in preparation, we can provide some of the results if needed]. For this reason, and previous research on the peripheral effects of centrally administered STZ, we believe the effects might exclusivelly affect the intestine, however, this remains to be confirmed in our future research.

We work with the STZ-icv model for a long time (MSP created the model in collaboration with prof. Hoyer) and we are aware that there are many biochemical changes in the brain at the 1 month time-point after the induction. Furthermore, many have been well documented in the literature (e.g. we briefly summarized the main pathophysiological effects of STZ-icv in line 70).  Behavioral changes were also most likely present (although not specifically tested in this cohort of animals, however we know, based on approximately 30 years of research with the model that 1 month after the STZ-icv administration cognitive deficit is a standard finding (tested with Morris Water Maze, Passive Avoidance, Novel Object Recognition, etc.). The specific design of this experiment is also not suitable for behavioral testing, as the animals were sacrificed 30 minutes after the antagonists were administered.

We recognize the importance of understanding intestinal changes in the context of pathophysiological changes taking place in the brain. Nevertheless, this is the first ever report on the intestinal changes in the model. We hope that future research (by us and others) will use this findings as a starting point for providing additional evidence that is needed to fully understand true biological backgroud of redox dyshomeostasis reported here. We didn't allocate a great proportion of the manuscript for STZ-icv-induced changes in the brain as this is already well-known, and would take away the focus from the main idea of the manuscript (pathophysiological events in the intestine), however if the reviewer considers this would greatly contribute to the readability of the manuscript we will be glad to extend the discussion of central events induced with STZ-icv at that time-point.

Introduction: well written. Please revise: “mendelian” should be “Mendelian.”

Corrected as suggested. Thank you.

Methods: page 4, line 162 statement regarding sample size should be mentioned in statistical section. It does not belong here.

We moved the information regarding the sample size in the methods section.

Data analysis: statement regarding the excluded samples (page 6, line 260) is not relevant. There is no need for these details as long as the n is described in figure legends or in some way or form.

We agree. We removed this information as it doesn't add important information to what was already stated.

Overall, the results section must be improved. Authors use the following terms to describe their findings “slight reduction,” “can reduce,” but I don’t see an objective description of the data from a statistical perspective. Are there true differences? There is no mention of p-values, statistical differences, etc. When you mention “slight reduction,” is this a trend? Values look the same to me due to the variability. Please revise.

Thank you for the suggestion. We altered the results section to describe the results more objectively. In both Fig 1 and Fig 2 we have reported raw data (scatter over the boxplot) to provide complete information we have obtained from the measurements. The data were then analyzed using two linear regression models for each variable. A „naive approach“ was used first testing the model without interaction to assess a general effect of both treatment 1 (STZ vs CTR) and treatment 2 (Exendin-3(9-39)amide vs. Saline). Point estimates with corresponsing 95% confidence intervals of the differences of group estimates (or geometric mean rations for log-transformed dependent variables) and p-values were reported in the main effects plot for each variable (e.g. Fig 1D-F for plasma; Fig 3 for raw data provided in Fig 2). Interaction terms for treatments were than included to test the hypothesis we were most interested in and provide an insight into whether the effects of inhibiting cerebral GLP-1 receptors differ between CTR and STZ animals (i.e. is the function of the brain-gut GLP-1 axis maintained in the STZ-icv rats)? Only the interaction term p-values were provided for the models including interaction terms (reported in each effect plot [vertical text]) as p-values from the models with the interactions terms do not provide reliable information and are not standardly reported when such models are used. Nevertheless, point estimates with corresponsing 95% confidence intervals of the differences of group estimates (or geometric mean rations for log-transformed dependent variables) have been reported to convey information on the effects observed. The effect plots provide complete information regarding comparisons of specific groups, and the „significance“ of each of the observed differences is evident from the plots. For models in which dependent variables were not log-transformed, estimate of 0 corresponds to „no difference“ as subtraction of two identical numbers gives 0. Analogously, for models with log-transfored dependent variables estimates of 1 indicate no difference (as division of two identical numbers gives 1). When 95% confidence interval of the point estimate of differences between two groups does not cross the red vertical line (0), 95% confidence intervals of estimates of two groups do not overlap (i.e. the groups come from different populations, or as usually interpreted the difference between groups is „statistically significant“). The same is true for contrasts from models with log-transformed dependent variables, however here the „no difference“ line goes through 1 instead of 0. We decided to present the results this way as all important parameters are communicated, and the reader can estimate true effects (instead of just focusing on p-values what usually leads to erroneous conclusions). Additionally, statistical outputs from all models have been provided in the Supplement 1 where all p-values are available as well.

My comments above pertain to the figures as well. My statistical knowledge and expertise are limited, but the way data are presented is hard to follow; why not just present the results from the whole model, p-values for the main effects and interaction can be presented separately in a table. None of those p-values highlighted here (figure 1 D-F) are presented in results in such terms.

Please see the previous answer. We reported the results in a way that is currently considered to be the best practice as it directly communicates treatment effects and the uncertainty in the estimates of these effects using an effects plot. Simply placing the * on the graphs fails to communicate what exactly was tested and how certain is the obtained result. Nevertheless, all information (in the table format) is already provided in the Supplement 1.

Page 8, line 321: “in contrast both LMWT and SH were reduced in the duodenal tissue of STZ-icv rats,”  compared to what?

We amended the sentence.

Page 8, line 326 are there only trends? Please revise.

The sentence was amended.

The authors failed to include the following in the discussion: how about inflammatory signaling? It is an important component in the pathophysiology of AD. How would you connect GI pathobiology to this aspect of  AD?

We agree that inflammatory signaling is an important mediator of GI dyshomeostasis-induced neurodegeneration. We included a brief discussion of the importance of GI-dyshomeostasis-induced systemic and neuro-inflammation in the discussion.

Page 14, line 463: authors talked about the role of oral galactose to restore plasma GLP-1 in the STZ-icv rats. If this is the case, how can you explain the galactose-induced models of neurodegeneration previously reported?

https://pubmed.ncbi.nlm.nih.gov/16555301/

This is an interesting and important question. Elucidation of neuroprotective effects of galactose is in focus of our group ever since the initial discovery that oral galactose treatment both prevents [1] and alleviates [3] cognitive deficit in STZ-icv-treated rats. We are well aware of the fact that parenteral galactose is standardly used for modeling oxidative stress and aging-related pathophysiological changes in rodents. In fact, the finding that oral galactose exerts neuroprotective effects while parenteral galactose induces oxidative stress was one of the reasons that led us to towards studying gut-related mechanisms that might be responsible for mediation of the protective effects of peroral galactose (GLP-1 secretion being one such gut-related mechanism [3]). As later findings indicated that in some cases peroral galactose might also be harmful [4,5] while parenteral galactose might exert cognitive enhancing effects in some specific scenarios[6], we believe current evidence speaks in favor of alternative explanations of the beneficial effects. For example, it is possible that intestinal mucosa buffers galactose enabling only tolerable amounts to reach CNS. In other words, bypassing the physiological barrier might be a key mechanism responsible for the detrimental effects because relatively high concentrations of galactose can reach the CNS this way challenging metabolic machinery. Even oral galactose given in a bolus dose via oral gavage might pose a „metabolic challenge“. In contrast, dissolution of galactose in the drinking water might enable a stable intake over a prolonged period allowing biochemical pathways to adapt to galactose utilization. We have discussed this here: [7].

There are of course many unknowns surrounding the effects of this important sugar. For example, mechanisms responsible for detrimental effects of galactose are still unknown and galactose oxidase, one of key enzymes that has been blamed for harmful effects of galactose in rodent models is a fungal enzyme that doesn't even exist in rodents [8] ! On the other hand, repeated findings of increased oxidative stress associated with galactose cannot be a mere coincidence. Nevertheless, it is possible that the findings of harmful effects do not necessary exclude the possibility of its beneficial action and it may even be the case that both detrimental and beneficial effects rely on the same biochemical pathways [7].

Although we find this extremely interesting, we believe expanding discussion on galactose might confuse the readers as it is not directly relevant to for presented findings

What is the physiological or pathophysiological relevance of this research design considering the focus is a chronic neurodegenerative disease like Alzheimer’s? This is not addressed in the manuscript.

The research design we used provides important insight in functioning of the GLP-1 brain-gut axis in STZ-icv rats 1 month after model induction. We believe this is highly relevant as i) STZ-icv rats standardly develop cognitive deficits 1 month after intracerebroventricular administration of the toxin, and researchers usually test different therapies in this time-point; ii) Our previous research indicated that GLP-1 might provide beneficial effects in the STZ-icv rats and novel findings related to the brain-gut axis may provide important information for understanding why this is so; iii) As hypothalamus is one of major targets for STZ-icv novel phenomenological findings related to the importance of brain incretin signaling for maintenance of the intestinal homeostasis provide information that might be important even for understanding early effects of STZ-icv.

The presented findings provide foundations for further research that may elucidate the etiopathogenesis of neurodegeneration in the STZ-icv model and brain-gut communication. Consequently, we consider the presented findings and the design to be highly relevant for the field.

  1. Salkovic-Petrisic, M.; Osmanovic-Barilar, J.; Knezovic, A.; Hoyer, S.; Mosetter, K.; Reutter, W. Long-Term Oral Galactose Treatment Prevents Cognitive Deficits in Male Wistar Rats Treated Intracerebroventricularly with Streptozotocin. Neuropharmacology 2014, 77, 68–80, doi:10.1016/j.neuropharm.2013.09.002.
  2. Salkovic-Petrisic, M. Oral Galactose Provides a Different Approach to Incretin-Based Therapy of Alzheimer’s Disease. J Neurol Neuromed 2018, 3, 101–107, doi:10.29245/2572.942X/2018/4.1204.
  3. Knezovic, A.; Osmanovic Barilar, J.; Babic, A.; Bagaric, R.; Farkas, V.; Riederer, P.; Salkovic-Petrisic, M. Glucagon-like Peptide-1 Mediates Effects of Oral Galactose in Streptozotocin-Induced Rat Model of Sporadic Alzheimer’s Disease. Neuropharmacology 2018, 135, 48–62, doi:10.1016/j.neuropharm.2018.02.027.
  4. Budni, J.; Pacheco, R.; da Silva, S.; Garcez, M.L.; Mina, F.; Bellettini-Santos, T.; de Medeiros, J.; Voss, B.C.; Steckert, A.V.; Valvassori, S. da S.; et al. Oral Administration of D-Galactose Induces Cognitive Impairments and Oxidative Damage in Rats. Behav Brain Res 2016, 302, 35–43, doi:10.1016/j.bbr.2015.12.041.
  5. Krzysztoforska, K.; Piechal, A.; Blecharz-Klin, K.; Pyrzanowska, J.; Joniec-Maciejak, I.; Mirowska-Guzel, D.; Widy-Tyszkiewicz, E. Administration of Protocatechuic Acid Affects Memory and Restores Hippocampal and Cortical Serotonin Turnover in Rat Model of Oral D-Galactose-Induced Memory Impairment. Behav Brain Res 2019, 368, 111896, doi:10.1016/j.bbr.2019.04.010.
  6. Chogtu, B.; Arivazhahan, A.; Kunder, S.K.; Tilak, A.; Sori, R.; Tripathy, A. Evaluation of Acute and Chronic Effects of D-Galactose on Memory and Learning in Wistar Rats. Clin Psychopharmacol Neurosci 2018, 16, 153–160, doi:10.9758/cpn.2018.16.2.153.
  7. Homolak, J.; Babic Perhoc, A.; Knezovic, A.; Kodvanj, I.; Virag, D.; Osmanovic Barilar, J.; Riederer, P.; Salkovic-Petrisic, M. Is Galactose a Hormetic Sugar? Evidence from Rat Hippocampal Redox Regulatory Network. bioRxiv 2021, doi:10.1101/2021.03.08.434370.
  8. Homolak, J.; Kodvanj, I.; Toljan, K.; Babić Perhoč, A.; Virag, D.; Osmanovic, J.; Mlinarić, Z.; Smailovic, U.; Trkulja, V.; Hackenberger, B.; et al. Separating Science from Science Fiction: A Non-Existent Enzyme Is a Primary Driver of Pathophysiological Processes in Galactose-Induced Rodent Models of Aging; 2020;

We thank the reviewer for the invested time and constructive comments that were provided. 

Sincerely,
Jan Homolak

Reviewer 2 Report

Review of Homolak et al “Failure of the brain glucagon-like peptide-1-mediated control of intestinal redox homeostasis in a rat model of sporadic Alzheimer’s disease”

Homolak et al aims assess the gastrointestinal homeostasis, the role of GLP1 in gastrointestinal redox and if the brain-gut GLP1 axis is function in a sporadic Alzheimer’s disease model. Th authors detail the assessment of plasma, duodenum and ileum for oxidative stress markers including TBARS, LMWT, SH, CAT and SOD levels. While the trends demonstrated by the data are interesting and do paint a picture of an involvement of the GLP1 in gastrointestinal redox homeostasis, more work is needed on the current manuscript prior to publication. Particularly, if the argument is that the brain/gut axis has a role in this regulation, more work is needed to demonstrate this phenomenon. Additionally, the current data in the manuscript needs further work to make it publication ready.

Major Concerns:

  • The introduction is extremely thorough in a literature review but takes away from the story of the data.
  • The data is difficult to interpret when comparing sections of figures across different figures. Rearrangement of figure panels is suggested to help readability.
  • Several trends seen in the data are that of just trends, were statistical analyses done on the data? If so, this needs to be represented on the graphs.
  • Variability of the samples within groups causes a reduced capacity to draw conclusions from the figures.
  • Discussion is written in a flow that is different than the paper, causing confusion on the main conclusions of the data.

Minor Concerns:

  • Several graphs are missing labels for X and Y axes including Figure 1D,E,F; 2O, R; Figure 3 in its enterity. Difficulty in determining the Y axes is of particular concern as it changes from 0 to 1 in several of the plots
  • The principal component analyses do not have defined what the shaded area represents.
  • Clarity of what the acronym terms such as TBARS, LMWT, SH, CAT and SOD are measuring is needed to aid in readability of the work.

Author Response

A point by point response to reviewers regarding „Failure of the brain glucagon-like peptide-1-mediated control of intestinal redox homeostasis in a rat model of sporadic Alzheimer’s disease“ by Homolak et al.

We wish to thank both the reviewers and the editors for their valuable time. We strongly the constructive comments will help us improve the manuscript.

Reviewer 2:

Review of Homolak et al “Failure of the brain glucagon-like peptide-1-mediated control of intestinal redox homeostasis in a rat model of sporadic Alzheimer’s disease”

Homolak et al aims assess the gastrointestinal homeostasis, the role of GLP1 in gastrointestinal redox and if the brain-gut GLP1 axis is function in a sporadic Alzheimer’s disease model. Th authors detail the assessment of plasma, duodenum and ileum for oxidative stress markers including TBARS, LMWT, SH, CAT and SOD levels. While the trends demonstrated by the data are interesting and do paint a picture of an involvement of the GLP1 in gastrointestinal redox homeostasis, more work is needed on the current manuscript prior to publication. Particularly, if the argument is that the brain/gut axis has a role in this regulation, more work is needed to demonstrate this phenomenon. Additionally, the current data in the manuscript needs further work to make it publication ready.

We thank the reviewer for time allocated for our manuscript and the constructive comments provided.

Major Concerns:

The introduction is extremely thorough in a literature review but takes away from the story of the data.

We agree with this observation. The main reason the introduction is thorough is beacuse brain-gut communication in the context of neurodegenerative diseases is a relatively novel topic. More importantly, this is the first manuscript on the brain-gut axis in the STZ-icv model, and we believe the STZ-icv model, the importance of the gastrointestinal system in neurodegeneration, and GLP-1 as an important novel therapeutic target in neurodegeneration (and an important link between gut and the brain) all had to be introduced to make the manuscript as informative as possible to the reader that might not be familiar with all of the above.

The data is difficult to interpret when comparing sections of figures across different figures. Rearrangement of figure panels is suggested to help readability.

Thank you for this suggestion, we rearranged the figures in a way that raw data and models are presented in the same figure. We believe this improves readability of the manuscript.

Several trends seen in the data are that of just trends, were statistical analyses done on the data? If so, this needs to be represented on the graphs.

We believe the reviewer raises an important point regarding statistical analysis and data presentation so we would like to provide an explanation of why we decided to report the data the way we did. We believe honest research should report „raw data“ so for all analyses we first reported raw values (scatter plot over the boxplots) (e.g. Fig 1A). The data were then analyzed using two linear regression models for each variable. A „naive approach“ was used first testing the model without interaction to assess a general effect of both treatment 1 (STZ vs CTR) and treatment 2 (Exendin-3(9-39)amide vs. Saline). Point estimates with corresponsing 95% confidence intervals of the differences of group estimates (or geometric mean rations for log-transformed dependent variables) and p-values were reported in the main effects plot for each variable (e.g. Fig 1D-F for plasma; Fig 3 for raw data provided in Fig 2). Interaction terms for treatments were than included to test the hypothesis we were most interested in and provide an insight into whether the effects of inhibiting cerebral GLP-1 receptors differ between CTR and STZ animals (i.e. is the function of the brain-gut GLP-1 axis maintained in the STZ-icv rats)? Only the interaction term p-values were provided for the models including interaction terms (reported in each effect plot [vertical text]) as p-values from the models with the interactions terms do not provide reliable information and are not standardly reported when such models are used. Nevertheless, point estimates with corresponsing 95% confidence intervals of the differences of group estimates (or geometric mean rations for log-transformed dependent variables) have been reported to convey information on the effects observed. The effect plots provide complete information regarding comparisons of specific groups, and the „significance“ of each of the observed differences is evident from the plots. For models in which dependent variables were not log-transformed, estimate of 0 corresponds to „no difference“ as subtraction of two identical numbers gives 0. Analogously, for models with log-transfored dependent variables estimates of 1 indicate no difference (as division of two identical numbers gives 1). When 95% confidence interval of the point estimate of differences between two groups does not cross the red vertical line (0), 95% confidence intervals of estimates of two groups do not overlap (i.e. the groups come from different populations, or as usually interpreted the difference between groups is „statistically significant“). The same is true for contrasts from models with log-transformed dependent variables, however here the „no difference“ line goes through 1 instead of 0. We decided to present the results this way as all important parameters are communicated, and the reader can estimate effects (instead of just focusing on p-values what usually leads to erroneous conclusions).

We reported the results in a way that is currently considered to be the best practice as it directly communicates treatment effects and the uncertainty in the estimates of these effects using an effects plot. All information critical for understanding the direction and size of each effect as well as the uncertainty can be inferred from the reported graphs.

Additionally, statistical outputs from all models have been provided in the Supplement 1.

Variability of the samples within groups causes a reduced capacity to draw conclusions from the figures.

There is often great variability in animal studies with relatively small number of animals per group. We also consider „removing outliers“, a process that is often encoutered in animal research a very bad and unjustified practice that rarely leads to elucidation of true biological phenomena. Consequently, there is a considerable within group variability in some cases.

Nevertheless, we don't see how the variability of the samples within groups causes a reduced capacity to draw conclusion from the results as all the results from the statistical models were „informed of the variability“ from the data. In other words, if the variability was not as great, 95% confidence intervals would have been narrower and some of the contrasts that now cross 0 (or 1 for log-transformed variables) would not cross the line, and would thus indicate „significant“ differences between the groups in cases where there is none at the moment due to the variability. As some of the variables demostrated considerable variability (as you fairly poited out), the estimates were assigned wider confidence intervals to inform the reader of the associated uncertainty. This was the primary reason we reported the results the way we did – to inform the reader of the effects (and „significances“), but also to communicate that some of the samples were „dispersed“ and that they therefore provided less certain estimates.

In other words, we completely agree that variability causes a reduced capacity to draw conclusion, however this has already been fully accounted for in all of the results we present.

Discussion is written in a flow that is different than the paper, causing confusion on the main conclusions of the data.

We did our best to structure the discussion in a way that it covers all the most important aspects of the results. This is the main reason the discussion was divided in 4 segments. So far we didn't receive criticism of the way the discussion was structured, however we are open for suggestions on how to further improve it to make the manuscript more readable.

Minor Concerns:

Several graphs are missing labels for X and Y axes including Figure 1D,E,F; 2O, R; Figure 3 in its enterity. Difficulty in determining the Y axes is of particular concern as it changes from 0 to 1 in several of the plots

The graphs in Fig 1D-F, Fig 2R,O and Fig 3 do not have missing labels. Model contrasts corresponding to differences between estimates of specific groups or ratios of the estimates for specific groups are designated on the lefthand side of the estimates with confidence intervals. For example, for a model with interaction term for non-log-transformed SOD (Fig 1D, under the title STZ moderates the effect of Ex-9) a point estimate with corresponding 95% confidence interval for difference between the control group and the STZ group treated with saline only was designated by „CTR SAL – STZ SAL“. For the log-transformed variables, the point estimates corresponding to „differences“ between the control group and the STZ group treated with saline only was designated by „CTR SAL/STZ SAL“ as the contrasts indicate geometric mean ratios.

As we explained earlier, the reason why the red line sometimes crosses 0 and sometimes 1 is due to the fact that subtraction of two identical numbers is equal to 0, while the ratio of two equal numbers is 1. As the contrasts of log-transformed variables give ratios of geometric means it is logical that the line would have to go through 1 instread of 0 for models with log-transformed dependent variables.

The principal component analyses do not have defined what the shaded area represents.

The shaded are reprenents 95% confidence ellipses around group baricentres. We included this information in the methods section.

Clarity of what the acronym terms such as TBARS, LMWT, SH, CAT and SOD are measuring is needed to aid in readability of the work.

We introduced the abbreviations in the methods section and in all figure legends to make it easier for the reader to interpret the figures.

We thank the reviewer for the invested time and constructive comments that were provided.

Sincerely,

Jan Homolak

Reviewer 3 Report

Comments attached

Round 2

Reviewer 1 Report

The authors were able to address most of my comments (ie. galactose discussion, physiological relevance). However, those replies were not necessarily reflected in the manuscript. If so, please highlight those sections or indicate pages/lines in the response to reviewers. Of note, there is no need to expand too much on the galactose discussion but one sentence should suffice and would be pertinent.

However, a major issue is the results section. The authors claim they altered the results section to describe their findings more objectively. I have reviewed this manuscript and this is not the case. No averages, raw values, p-values, or trends are "objectively" (using numbers) described in the section. This section is merely descriptive. The authors claim that the best way to present their data is using an effects plot but in general, all figures are hard to follow and the results do not make the work any easier. I am not sure why going through all these different levels of statistical analysis and not reporting them adequately in results. Please reconsider the way you are presenting your results. It is taking away the relevance of your findings. 

Author Response

We wish to thank the reviewer for their constructive comments.

The authors were able to address most of my comments (ie. galactose discussion, physiological relevance). However, those replies were not necessarily reflected in the manuscript. If so, please highlight those sections or indicate pages/lines in the response to reviewers. Of note, there is no need to expand too much on the galactose discussion but one sentence should suffice and would be pertinent.

Thank you for your suggestions. The pathophysiological relevance of the STZ-icv model has been presented in the introduction (67-77). In lines 77-82 we included an additional sentence emphasizing that STZ-icv rats usually develop AD-like pathophysiological alterations quite rapidly (most have been observed already 1 month after intracerebroventricular administration). This is also emphasized in the referenced articles (e.g. [22] Optimization of Intracerebroventricular Streptozotocin Dose for the Induction of Neuroinflammation and Memory Impairments in Rats by Ghosh et al.).

Additional information regarding galactose treatment and the fact that it is often used for modeling aging in rodents have been added in lines 149-157. 

Thank you for the suggestion. We now included relevant contrast information from both the main effects models (testing overall general effects where suitable) and the interaction models for specific treatment 2 differences at the levels of treatment 1 in the text in the Results section.

Thank you for your suggestions once again.

Sincerely, 
Jan Homolak (for the authors)

Reviewer 2 Report

the authors provided sufficient reasoning and rationale in the rebuttal letter

Author Response

The authors provided sufficient reasoning and rationale in the rebuttal letter

We wish to thank the reviewer for their constructive comments. 

Sincerely, 
Jan Homolak (for the authors)